Copepods enhance nutritional status, growth and development in Atlantic cod (Gadus morhua L.) larvae — can we identify the underlying factors?

Karlsen Ørjan 1 2 OrjanK@imr.no
van der Meeren Terje 1 2
Rønnestad Ivar 3
Mangor-Jensen Anders 1
Galloway Trina F. 4
Kjørsvik Elin 5
Hamre Kristin 6
1 Institute of Marine Research, Austevoll Research Station , Storebø , Norway
2 Hjort Centre for Marine Ecosystem Dynamics, Institute of Marine Research , Bergen , Norway
3 Department of Biology, University of Bergen , Bergen , Norway
4 SINTEF Fisheries and Aquaculture , Trondheim , Norway
5 Department of Biology, Norwegian University of Science and Technology , Trondheim , Norway
6 National Institute of Nutrition and Seafood Research , Nordnes, Bergen , Norway
Esteban María Ángeles
Electronic publication date: 2015 May 19
Publication date: 2015
Volume: 3
Electronic Location ID: e902
Received 2014 Dec 28; Accepted 2015 Mar 31
Copyright: © 2015 Karlsen et al.
Copyright year: 2015
Copyright holder: Karlsen et al.
License: This is an open access article distributed under the terms of the Creative Commons Attribution License, which permits unrestricted use, distribution, reproduction and adaptation in any medium and for any purpose provided that it is properly attributed. For attribution, the original author(s), title, publication source (PeerJ) and either DOI or URL of the article must be cited.
License URL: https://creativecommons.org/licenses/by/4.0/

Keywords: Atlantic cod, Start feeding, Nutrition, Natural zooplankton, Copepods, Larvae, Taurine, Gadus morhua, Rotifers, Artemia

Funding: Ministry of Fisheries and Coastal Affairs Research Council of Norway The work was funded by the Ministry of Fisheries and Coastal Affairs and the Research Council of Norway (CODE knowledge platform project; Grant no. 199482/S40). The funders had no role in study design, data collection and analysis, decision to publish, or preparation of the manuscript.

==============================
The current commercial production protocols for Atlantic cod depend on enriched rotifers and Artemia during first-feeding, but development and growth remain inferior to fish fed natural zooplankton. Two experiments were conducted in order to identify the underlying factors for this phenomenon. In the first experiment (Exp-1), groups of cod larvae were fed either (a) natural zooplankton, mainly copepods, increasing the size of prey as the larvae grew or (b) enriched rotifers followed by Artemia (the intensive group). In the second experiment (Exp-2), two groups of larvae were fed as in Exp-1, while a third group was fed copepod nauplii (approximately the size of rotifers) throughout the larval stage. In both experiments, growth was not significantly different between the groups during the first three weeks after hatching, but from the last part of the rotifer feeding period and onwards, the growth of the larvae fed copepods was higher than that of the intensive group. In Exp-2, the growth was similar between the two copepod groups during the expeimental period, indicating that nutrient composition, not prey size caused the better growth on copepods. Analyses of the prey showed that total fatty acid composition and the ratio of phospholipids to total lipids was slightly different in the prey organisms, and that protein, taurine, astaxanthin and zinc were lower on a dry weight basis in rotifers than in copepods. Other measured nutrients as DHA, all analysed vitamins, manganese, copper and selenium were similar or higher in the rotifers. When compared to the present knowledge on nutrient requirements, protein and taurine appeared to be the most likely limiting nutrients for growth in cod larvae fed rotifers and Artemia. Larvae fed rotifers/Artemia had a higher whole body lipid content than larvae fed copepods at the end of the experiment (stage 5) after the fish had been fed the same formulated diet for approximately 2 weeks.

Introduction

In aquaculture, the production of Atlantic cod (Gadus morhua) juveniles is based on indoor intensive systems with start-feeding tanks supplied with temperature-controlled seawater and in which the larvae are fed enriched rotifers (Brachionus spp.) at the onset of exogenous feeding. This may be followed by a period of feeding with enriched brine shrimp (Artemia salinas) before the larvae are weaned onto formulated feed. However, the quality of the cod juveniles achieved with this production method is variable and often suboptimal. The proportion of deformed fish is often higher than when the larvae are fed natural zooplankton that consists mainly of copepods (Fjelldal et al., 2009; Imsland et al., 2006). Moreover, the rate of growth of larvae fed rotifers/Artemia is lower than that of larvae fed copepods (Busch et al., 2010; Busch et al., 2011; Evjemo, Reitan & Olsen, 2003; Hamre et al., 2013; Imsland et al., 2006; Koedijk et al., 2010; Liu & Xu, 2009). Analyses of stomach content from wild caught cod larvae show that an assemblage of various species and stages of copepods are the most important food items for cod larvae in their natural habitat (Wiborg, 1948; Last, 1978).

It has been hypothesised that the consistent and inferior growth rates and juvenile quality of cod fed rotifers/Artemia are due to suboptimal nutrition, indicated by numerous differences in the nutrient content of rotifers and copepods (Hamre et al., 2008a; Hamre et al., 2013; Imsland et al., 2006; Srivastava et al., 2006; van der Meeren et al., 2008; Oie et al., 2015, unpublished data). The dry-weight concentration of protein is much higher in copepods than in rotifers, and such levels cannot easily be obtained by enrichment of the rotifers (Hamre et al., 2013; Srivastava et al., 2006). Levels of essential n-3 fatty acids and phospholipids are often lower in rotifers than in copepods (van der Meeren et al., 2008). However, this can be improved by using high levels of n-3 fatty acids in the culture medium and by culturing lean rotifers which contain low levels of triacylglycerol and thus have a higher ratio of phospholipids (PL) to total lipids (TL) (Hamre et al., 2013). It is well known that raising the ratio of PL to TL in larval feeds enhances growth (Cahu, Zambonino Infante & Barbosa, 2003). Commercial rotifer diets usually contain sufficient levels of vitamins to meet the requirements of marine fish larvae, with the possible exception of vitamin D and K, on which there are few studies (Hamre et al., 2013). However, it is possible that microminerals are present in suboptimal concentrations (Hamre et al., 2008a), probably underlying part of the reduced survival of larvae (Hamre et al., 2008b). To rectify these problems, new enrichment protocols have been developed for some of the potentially deficient nutrients (Nordgreen, Penglase & Hamre, 2013; Penglase et al., 2010; Srivastava et al., 2012; Srivastava, Stoss & Hamre, 2011), which enable comparable levels in rotifers as those observed in copepods to be reached. Macrominerals and most of the B-vitamins are present in rotifers and Artemia at levels comparable to those observed in copepods (Hamre et al., 2013) and therefore seem to be sufficient, on the basis of our current knowledge. It should be noted that the rotifers’ and Artemia’s own metabolisms prevent the levels of some nutrients from being customised or raised by enrichment procedures.

In addition to nutrient composition, the size and energy content of rotifers and Artemia may also limit fish growth. Marine fish larvae select prey of increasing size as they grow if different prey sizes are available (Hunter, 1980; Kuhlman, Quantz & Witt, 1981; Last, 1978; Olsen et al., 2000; Stepien, 1976; van der Meeren, 1991). Two groups of cod larvae that were fed small or large copepods were similar in size until 25 days post hatch (dph), but thereafter growth was faster in the larvae fed large copepods (Busch et al., 2011). Furthermore, feeding cod larvae large rotifers instead of small ones, with similar prey biomass per tank, resulted in significantly larger larval size after just a few days of feeding (G Øie, SINTEF Fisheries and Aquaculture, pers. comm., 2014). It is therefore possible that the methods most often employed in feeding copepods to cod, e.g., increasing prey size as the larvae grow, supplies more energy to the larvae than when they are fed small rotifers and Artemia, and that different prey size is the reason for the observed differences in growth rates. Another possible explanation for these variations in growth are differences in feed intake caused by sensory stimuli, due to species-specific prey movements or release of chemically mediated attractants (van der Meeren, 1991; Yacoob, Browman & Jensen, 2004).

The purpose of the present study was to identify the underlying mechanisms that lead to differences in the growth and development between cod larvae fed rotifers/Artemia and those fed copepods. We used state-of-the-art cultivation and enrichment methods for rotifers and Artemia, while copepods were harvested from a local semi-controlled seawater pond system (Naas, van der Meeren & Aksnes, 1991; van der Meeren et al., 2014). We conducted two experiments. In the first experiment (Exp-1), two groups of cod larvae in triplicate tanks were fed rotifers/Artemia or copepods. Here we present data that cover larval growth and survival and the nutritional status of the prey types and the developing cod larvae. Additional results describing the ontogeny of cod larvae, data for transcriptomics, microRNA-sequencing and metabolomics, and a wide range of biological processes in the larvae will be reported elsewhere.

The second trial (Exp-2) was designed to distinguish between prey size and prey nutrient composition as the cause of the enhanced growth of cod larvae fed copepods. Two groups of larvae were subjected to the same feeding regimes as in Exp-1, i.e., one with increasing copepod size (from nauplii to copepodids) as the larvae grew, and another with rotifers from the onset of first-feeding followed by Artemia. Additionally, a third group of larvae were fed small prey size (copepod nauplii) throughout the whole experiment. We hypothesised that similar growth rates in larvae fed small and increasing sized copepods respectively, indicate that nutritional differences between rotifers/Artemia and copepods is the main reason for differences in larval growth.

Material and Methods

Fish material and experimental design

This study was carried out within the Norwegian animal welfare act guidelines, in accordance with the Animal Welfare Act of 20th December 1974, amended 19th June, 2009, at a facility with permission to conduct experiments on fish (code 93) provided by the Norwegian Animal Research Authority (FDU, www.fdu.no). The first trial was assumed to be a nutrition trial not expected to harm the animals, no specific permit was required under the guidelines. The second experiment was approved by FDU, FOOTS ID 5448.

Fertilized eggs for Exp-1 were obtained from Atlantic cod broodstock of coastal origin, western Norway, which were held at the Institute of Marine Research (IMR), Austevoll Research Station. The naturally spawning fish, 45 females and 25 males, were kept in a tank of seawater (yearly means; temperature: 7.9±0.3 °C, salinity: 34.7±0.2 g L−1). The eggs were collected overnight March 15, 2012, using an egg collector as described by van der Meeren & Lønøy (1998). Mean egg diameter was 1.29 mm, and fertilization rate was 79%.

Eggs for Exp-2 were obtained from siblings of the broodstock used in Exp-1, kept at sea in net pens at the IMR facility in Parisvannet, Øygarden. Eggs were stripped on April 4, 2013 from three females (1.2 L in total), and fertilized with mixed milt from 10 males. Sea temperature at time of stripping was 4.0 °C. The fertilized eggs were immediately transported for about three hours by car to IMR-Austevoll in a 15 l closed plastic bag with an air space, held in a polystyrene box. On arrival at Austevoll, the temperature had risen to 4.7 °C. The eggs were disinfected using 400 mg L−1 glutaraldehyde for 8 min according to Harboe, Huse & Øie (1994). Mean egg diameter was 1.30 mm, and fertilization rate was 86%.

Water for the Austevoll station is pumped from a depth of 168 m, sand-filtered, temperature-adjusted and aerated before it enters the tanks. The eggs were held in 70 L black polyethylene incubators with conical bases; 1 L of eggs per tank in Exp-1, and 0.20 L in Exp-2. The tanks had gentle aeration (van der Meeren & Lønøy, 1998), water supplied at 0.5 L min−1, 5.8–6.1 °C, and with continuous light. Dead eggs were removed daily and measured volumetrically. In Exp-1, 50% hatching (day 0) was reached on 8th of April while in Exp-2, 50% hatching on the 18th of April. In both experiments, cod larvae were transferred to start-feeding tanks on 4 dph.

In Exp-1, 50,000 cod larvae were stocked in each of the six black PEH 500 L start-feeding tanks with gentle aeration, in order to prevent feed organisms and larvae from clogging the sieves. Larvae in three of the tanks (randomized) were fed copepods, while in the other three tanks they were first-fed enriched rotifers followed by enriched Artemia. Water flow was increased with age, from 1 L min−1 initially to 6 L⋅ min−1 at weaning, and further to 10 L min−1 at experiment termination, giving a water exchange rate of 32 tank volumes day−1 (van der Meeren et al., 2011). The temperature was gradually increased from 8 °C at transfer to 11.6 °C at 11 dph (Fig. 1), where it was kept for the rest of the experiment. Salinity was constant at 34.7 ± 0.2 g L−1. The tanks were equipped with air skimmers to remove any biofilm on the water surface. An automated cleaning arm (van der Meeren et al., 1998), rinsed the bottom of the tanks as needed from 12 dph. To modify the visual feeding environment and enhance feeding (Naas, Næss & Harboe, 1992; van der Meeren, Mangor-Jensen & Pickova, 2007), 10 ml algal paste (Marine microalgae concentrate, Nannochloropsis sp., Instant Algae®, Nanno 3600; Reed Mariculture, Campbell, California, USA) was added to each tank 15 min prior to each meal until 36 dph (Fig. 2). A 16L:8D photoperiod was employed, with 30 min simulated twilight at each dusk and dawn. The light source consisted of two 20 W tungsten halogen light bulbs (12 V) over each tank that provided 300–500 µW/cm2 at the water surface (IL 1400A photometer; International Light Inc., Boston, Massachusetts, USA).

Figure 1 Physical conditions of experiments.

Figure 1. Water temperature, relative water exchange rate given as tank volume day−1, and oxygen saturation during the experimental period (4–90 dph) for the copepod and the rotifer/ Artemia groups in both experiments (Exp-1 and Exp-2).

Figure 2 Feeding regime in Exp-1.

Feeding regimes, sampling (black triangles), and use of “green water” (where an algal paste is added to the water) in Exp-1; (A) larvae fed rotifers and Artemia, (B) copepod feeding regime. Copepod size fractions are given between the vertical dotted lines. Particle diameter is given for formulated feed: AgloNorse (AN) and Gemma Diamond (GD).

In Exp-2, 1900 cod larvae were distributed randomly to each of 12 green PEH 50 L startfeeding tanks at 4 dph. Three feeding regimes were employed: larvae in four tanks received enriched rotifers followed by enriched Artemia, another four tanks received copepod nauplii and copepodids with prey whose size increased with larval age, and the last four tanks were given copepod nauplii without increases in prey size through the larval period (Fig. 3).

Figure 3 Feeding regime in Exp-2.

Feeding regimes, sampling (black triangles), and use of green water in Exp-2; (A) feeding regime with rotifers and Artemia, (B) copepod feeding regime with increasing prey size, and (C) copepod feeding regime with small prey size. Copepod size fractions are given between the vertical dotted lines.

The tank design and setup employed in Exp-2 were miniatures of the tanks in Exp-1, but without cleaning arm and surface skimming. Cleaning was done manually by siphoning the bottom when necessary. Water quality was similar to Exp-1, except that the water was not temperature-adjusted. The temperature therefore ranged between 7.1 and 8.6 °C (Fig. 1). The water flow was kept stable at 0.42 L min−1 during the experiment. Oxygen saturation was not measured as the daily water-exchange rate was high relative to tank size (Fig. 1). Illumination was provided by broad-spectrum fluorescent light tubes (Philips 965 TL-D 90 De Luxe Pro; Philips, Amsterdam, Netherlands) providing 200–300 µW/cm2 at the water surface. Algal paste (3.3 mL tank−1, Nannochloropsis sp.; Reed Mariculture, Campbell, California, USA) was added 15 min prior to every feeding from 4 to 33 dph (Fig. 3).

Larval feed and feeding regimes

In Exp-1, the cod larvae were fed three times a day (09:45, 15:15 and 19:00). The rotifer cultures were seven-day-old batch cultures, kept in 2 m3 conical tanks at 24 °C at densities ranging between 850 and 2600 rotifers mL−1, and fed four times h−1. The diet consisted of dry baker’s yeast (0.11–0.18 g million rotifers−1) and Rotifer Diet ® (0.3–1.5 g million rotifers−1; Reed Mariculture Inc., Calfornia, USA). The rotifers were enriched before they were fed to the larvae. The enrichment protocol was performed daily by moving the desired number of rotifers into an enrichment tank at densities in the range of 1000–2000 rotifers ml−1. The culture was given 0.6 mg Sel-Plex® (Alltech, Vejle, Denmark) and 0.2 g Larviva Mulitigain (Biomar, Trondheim, Norway) per million rotifers over a 20 min period each hour from 12:00 to 08:00 the following day. After enrichment the culture was rinsed in clean 24 °C seawater, then cooled to the same temperature as the start-feeding tanks, and finally stored under moderate aeration and continued cooling until given to the cod larvae. From 32 to 35 dph the rotifers were gradually replaced with enriched Artemia, which were fed until 63 dph. The Artemia were hatched from SepArt cysts (INVE Aquaculture, Dendermonde, Belgium), and enriched at densities of maximum 500 Artemia mL−1 using 0.2 g Larviva multigain L−1 tank volume, given four times from 20:00 to 08:00. After enrichment, the Artemia were rinsed in clean seawater, cooled, and stored like the rotifers. From 58 to 63 dph, Artemia were fed to the cod larvae only once a day at 15:15. Weaning to the formulated diet AgloNorse ® 400–600 µm (Tromsø Fiskeindustri AS, Tromsø, Norway) started by hand feeding on 55 dph, and continued using feeding automats from 57 dph (Fig. 2A).

Copepods were collected from “Svartatjern,” a nearby 25 000 m3 sea-water pond system (Naas, van der Meeren & Aksnes, 1991). Pond operation, hydrographical and biological monitoring, and copepod filtration system and harvest procedures are described in detail by van der Meeren et al. (2014). A UNIK-900 wheel filter (Unik Filtersystem AS, Os, Norway) was used to fraction, concentrate and harvest the copepods. The filter fractions used in Exp-1 (Fig. 2B) were 80–150 µm (4–11 dph), 80–180 µm (11–18 dph), 80–212 µm (18–23 dph), 80–250 µm (23–36 dph), and 80–350 µm (37–44 dph). The collected copepods were concentrated under aeration by a 80 µm plankton net and transported in a 10 L bucket with lid and aeration for 7 min by boat to the larval rearing facility. On arrival, the concentrated mixture of copepods was diluted to 30–60 L with 7–8 °C seawater and stored under the same conditions as the enriched and cooled rotifers and Artemia. Viability was checked under a 40× magnification binocular microscope. Copepods were harvested from the pond one to three times a day, depending on larval needs and plankton availability in the pond. The quantities of copepods added to the larval tanks are shown in Fig. 2B. The larvae were weaned to a formulated diet from 37 dph as described above. At 70 dph, AgloNorse was switched to Gemma Diamond® 1.0 (Skretting, Stavanger, Norway).

In Exp-2, larvae in four tanks were fed rotifers and Artemia, which were produced as in Exp-1. Rotifers were used from 4 to 34 dph, while Artemia were used from 34 to 47 dph (Fig. 3A). The copepods were also obtained as described in Exp-1. The four tanks with larvae fed copepods which increased in size with larval age were given the 80–150 µm fraction from 4 to 15 dph, 80–180 µm from 16 to 27 dph, 80–212 µm on 28 to 34 dph, and 212–250 µm from 35 dph and onwards (Fig. 3B). Finally, larvae in the four tanks given small sized prey only were fed copepod nauplii collected from the 80–150 µm fraction from 4 to 15 dph and 80–180 µm for the rest of the experiment (Fig. 3C). Exp-2 was terminated on 47 dph. To measure copepod size, samples were fixed in 1:50 Lugol’s solution. Samples taken in the morning and afternoon on four days were photographed with two images per sample (Leica MS5 with an Olympus DP70), and the total length of all nauplii and the prosome length of all copepodids on the images were measured using ImageJ (v2013-2, National Institute of Health, Bethesda, Maryland, USA).

Larval sampling and staging

Due to the unequal growth of the larvae in the different dietary groups, the dates of sampling in Exp-1 were adjusted according to larval size rather than age. Before the trial, a system of stages was developed, which was based on the sequence of mineralisation of craniofacial bones and is correlated with certain ranges of standard lengths (Dr. Ø Sæle, NIFES, pers. comm., 2012). Here we report data from larval stages 0–5 and juveniles. Sampling at stages 0, 1, and 2 corresponds to 4, 11 and 22 dph for both groups since the growth rate was similar in the early stages. For stages 3, 4, 5 and juveniles the time for sampling corresponds to 28, 36, 52 and 73 dph in the copepod groups and 30, 53, 70 and 84 dph in the rotifer groups (Table 1). Additional samples for growth measurements were collected in between these ages. The samples were collected two hours after the first meal of the day in order to standardise gut filling conditions. The reason for sampling larvae with a full gut was that parallel samples were taken for studies of digestive physiology, the results of which are reported elsewhere. The time of feeding prior to sampling was adjusted to keep a fixed standardized time between feeding and sampling for all tanks. Larvae used for length and dry weight analysis were individually sedated, photographed, killed with an overdose of MS 222, rinsed in filtered distilled water, laid in pre-weighed aluminium beakers, and freeze-dried. The beakers with larvae were weighed, and larval weight determined by subtraction. Larval standard length, from the tip of the snout to the end of the notochord, was measured on the photographs. At the two last samplings for each of the groups, larval length and wet weight were measured. Larvae used for analysis of nutrient composition were sampled at stages 3 (at the end of the rotifer period) and 5 (both groups were weaned after sampling at stage 4). Pooled larvae (one sample per tank) were killed by an overdose of MS 222 and sieved through a plankton filter which was subsequently patted dry from underneath with a paper towel. Samples were distributed to tubes for the different nutrient analyses, immediately frozen on dry ice, and then transported to National Institute of Nutrition and Seafood Research (NIFES) and stored at −80 °C until analyses.

Table 1 Overview of the sampling in Exp-1.

Overview of larval stages at sampling in Exp-1, corresponding larval age and standard length (SL, mean ±SD) and approximate number of larvae sampled per tank (No). Superscript letters indicate significant differences between treatments in SL (t-test, p < 0.05). Larvae were fed either copepods or rotifers in stages 0–3 and Artemia in stages 3–4 (Rotifers). Both groups were weaned after the sampling at stage 4. Stages were described by Ø Sæle et al. (2012, unpublished data) and correlated with those given by Hunt von Herbing et al. (1996).

Stage	Age (dph)	SL (mm)	No	
	Rotifers	Copepods	Rotifers	Copepods	Rotifers	Copepods	
0	4	4	4.5 ± 0.2	4.5 ± 0.2	3,350	5,300	
1	11	11	5.2 ± 0.4	5.1 ± 0.3	3,850	3,400	
2	22	22	6.9 ± 0.6	7.0 ± 0.7	2,180	1,800	
3	31	29	8.5 ± 0.7	10.1 ± 0.9	1,550	2,800	
4	54	37	14.8 ± 2.6	12.9 ± 1.4	1,435	1,300	
5	71	53	26.5 ± 3.2a	23.7 ± 3.1b	1,360	1,300	
Juvenile	85	74	48.8 ± 6.6	48.9 ± 5.8	1,350	1,200	

Table 2 Analytical methods employed.

Analytical methods for the different nutrients.

Analyte	Principle	Reference	
Dry matter	Gravimetric after freeze drying	Hamre & Mangor-Jensen, 2006	
Protein	N × 6.25 Leco N Analyzer	Hamre & Mangor-Jensen, 2006	
Taurine	Total amino acids	Espe et al., 2006	
Fatty acids	Transmethylation extraction and GC/FID	Lie & Lambertsen, 1991	
Lipid classes	HPTLC	Jordal, Lie & Torstensen, 2007	
Vitamin C	HPLC	Mæland & Waagbø, 1998	
Vitamin A	HPLC	Moren, Næss & Hamre, 2002	
Vitamin D	HPLC	CEN , 1999	
Vitamin E	HPLC	Hamre, Kolås & Sandnes, 2010	
Sum vitamin K3	HPLC	CEN , 2003	
Carotenoids	HPLC	Ørnsrud et al., 2004	
Microminerals	ICP-MS	Julshamn et al., 2004	
Iodine	ICPMS	Julshamn, Dahl & Eckhoff, 2001	

Exp-2 was basically a growth and survival study, and larval samples were taken for weight and length only, using the protocol described above. The larvae were not staged and the samples were taken at the same ages for all groups.

Sampling of live prey for chemical analyses

In Exp-1, three samples of enriched rotifers and two samples of enriched Artemia were taken for analysis of nutritional composition. The samples were taken from the cooling tanks, washed in fresh water, sieved and patted dry from underneath the sieve with a paper towel, frozen and stored at −80 °C until analysed. The samples were filtered on a 60 µm plankton net, patted dry with a paper towel, frozen immediately at −80 °C and stored until analysis.

In Exp-2, rotifers were sampled on 13 and 27 dph, Artemia just after the experiment ended, and small and large copepods just before first-feeding and at 6 and 21 dph.

Biochemical analysis

The nutrient composition of fish and diets was measured by ISO-certified routine methods at NIFES. Table 2 presents an overview over the biochemical methods with analysis principles and references. Protein is given as N × 6.25 for cod larvae, N × 4.46 for rotifers and N × 5.30 for Artemia and copepods, respectively (Hamre et al., 2013).

Data analysis and statistical analysis

Specific growth rate (SGR) and daily length increment (DLI) were calculated according to Ricker (1979):

SGR (% day−1) = 100(eg − 1), where g = (lnW2 − lnW1)(t2 − t1)−1

DLI (mm day−1) = (L2 − L1)(t2 − t1)−1

W2 and W1 are dry weights, while L2 and L1 are lengths at times t2 and t1, respectively.

Differences in survival between the two treatments in Exp-1 were compared using Mann–Whitney U test, while the three treatments in Exp-2 were compared using Kruskal–Wallis ANOVA. Size (length, weight) was compared using nested two-way ANOVA (treatment, time) with tanks nested under treatment, followed by a Tukey HSD test if significant if p < 0.05. Growth rates (SGR and DLI) were compared for separate stages. In Exp-1 compared by means of Student’s t-test, and in Exp-2 by a one-way ANOVA and if significant (p < 0.05) followed by Tukey HSD.

The data on the nutrient composition of the feeds were first checked for homogeneous variances using Levene’s test and log transformed if significant, using the Statistica software package (ver. 12, StatSoft Inc. Tulsa, Oklahoma, USA). They were then analysed by ANOVA and Tukey HSD test for unequal sample sizes. Nutrient composition of larvae in Exp-1 on stages 3 and 5 were analysed by t-tests.

Results

Survival

In Exp-1, between 4223 and 6210 larvae were collected from each of the tanks at the end of the trial. Estimates of survival, calculated as the proportion of the larvae counted out of the tanks at termination, divided by the number of larvae added to the tanks at start (after sampling at 4 dph, N = 50 000) corrected for samplings (N = 10 000), ranged between 11 and 16%. Mean (± SD) survivals were 14 ± 2% and 12 ± 1% in the copepod and the rotifer/Artemia fed groups, respectively, and were not significantly different between the treatments (Mann–Whitney U-test, p = 0.190).

Estimated survival at the end of Exp-2, calculated as above, ranged from 10 to 47%. Mean survival was lower in the group fed rotifers/Artemia (13%, range 10–20%) than in the small copepod (39%, range 35–47%) and large copepod (37%, range 31–43%) groups (Kruskal–Wallis ANOVA, p < 0.001). The large and small copepod groups did not differ (Mann–Whitney U-test, p = 0.837).

Growth

In Exp-1 the larvae were 4.13 ± 0.12 mm (N = 25) at hatching, and 4.46 ± 0.18 mm (N = 24) at 4 dph (Stage 0). The growth in weight and length, DLI and SGR of larvae in Exp-1 were not significantly different between the groups at stages 0–2, from 4 to 22 dph (Figs. 5A, 5C and Table 3A). The copepod group displayed significantly improved DLI and SGR compared to the intensive group between stages 2 and 4 (t-test, p < 0.01), while there were no significant differences from stages 4 to 5 or 5 to juvenile in length increment or growth rate (Figs. 5A, 5C and Table 3A)

Figure 4 Copepod size fractions.

Mean prosome (copepodids) or total length (nauplii), quartiles (box) and range (lines) of nauplii (open) and copepodids (grey) in the different filter fractions used as feed in the experiments. No nauplii were present in the 180–250 µm or 212–250 µm fractions.

Figure 5 Growth in length and dry weight in both experiments.

Growth in length (mm) and dry weight (mg) in Exp-1 (A and C) and Exp-2 (B and D) of cod larvae fed either rotifers and Artemia or copepods. In Exp-1, larvae were fed copepods of increasing size. Stages (St. 0–5 and Juveniles, see text for explanation) are indicated. In Exp-2, one group was fed copepod nauplii and another group copepods of increasing size. Note logarithmic scale in C and D.

Table 3 Larval growth rates.

Daily length increment (DLI, mm day−1) and specific growth rate (SGR, weight %day−1) of cod larvae fed rotifers and Artemia (Rotifers) or copepods during the start-feeding period. Superscripts indicate significant differences in tank means during the periods (Exp-1 t-test; Exp-2 one way ANOVA and Tukey HSD, p < 0.05). (A) Exp-1, n = 3, calculations based on stages; (B) Exp-2, n = 4, calculations based on age. Data are given as mean ± SD.

(A) Stages	DLI	SGR	
(Age as dph, rot/cop)	Rotifers	Copepod	Rotifers	Copepod	
0–2 (4–22/4–22)	0.13 ± 0.03	0.14 ± 0.02	11.9 ± 1.0	11.9 ± 1.6	
2–4 (22–36/22–53)	0.25 ± 0.05a	0.42 ± 0.06b	8.6 ± 0.4a	15.5 ± 2.5b	
4–5 (36–52/53–70)	0.73 ± 0.04	0.67 ± 0.01	11.6 ± 0.8	12.9 ± 0.7	
5-Juvenile (52–73/70–84)	1.58 ± 0.29	1.25 ± 0.05	12.0 ± 1.9	11.4 ± 0.4	
(B) Period	DLI	SGR	
(dph)	Rotifers	Small cop.	Large cop.	Rotifers	Small cop.	Large cop.	
4–20	0.08 ± 0.02a	0.10 ± 0.01a	0.08 ± 0.01a	4.74 ± 0.69a	5.46 ± 0.57a	4.75 ± 0.43a	
20–34	0.16 ± 0.05a	0.21 ± 0.02a,b	0.26 ± 0.02b	8.93 ± 2.15a	12.0 ± 0.6b	13.8 ± 1.0b	
34–47	0.11 ± 0.02a	0.35 ± 0.04b	0.35 ± 0.04b	4.44 ± 1.08a	10.4 ± 1.1b	10.4 ± 1.8b	

In Exp-2, the larval dry weight at 4 dph was 0.083 ± 0.024 mg and length 4.51 ± 0.21 mm (Figs. 5B and 5D). Mean length and weight did not differ between the treatments until 20 dph (nested ANOVA, p > 0.935). Thereafter, both copepod groups were significantly longer and heavier than the intensive group (nested ANOVA, p < 0.005). No significant differences in length were observed between larvae of the two copepod treatments (nested ANOVA, p > 0.398) except at 40 dph, when the group fed large copepods was significantly longer than the group fed small copepods (p < 0.001). Similarly, there were no significant difference in dry weights between the two copepod groups except at days 40 and 47, where the larvae fed the large copepods had higher dry weights than those fed the small copepods (p < 0.015, Figs. 5B and 5D).

The DLI and SGR in Exp-2 were not significantly different between the treatments from 4 to 20 dph (Table 3B). Thereafter (days 20–34 dph and days 34–47 dph), growth was slower in the intensive group, particularly during the last period, when the SGR of the intensive group fell to only half of the previous period (from 8.9 to 4.4% day−1) (Table 3B). There were no significant differences in DLI or SGR between the two copepod groups (Mann–Whitney U-test, p > 0.15).

Prey size and nutrient composition of live feed organisms and cod larvae

The nauplii had overall lengths of 60 to 225 µm, while the copepodids had prosome lengths of 225 to 660 µm with little overlap (Fig. 4). Rotifers typically have a lorica length of 100–200 µm and Artemia instar I about 4–600 µm.

The nutrients present at lower levels in rotifers than in copepods in Exp-1 were protein (p < 0.001), taurin (p < 0.001), astaxanthin (p < 0.001), iodine (p < 0.05) and zinc (p < 0.001) (Table 4). There were also differences in fatty acid composition, in that the rotifers’ ARA fraction (p < 0.0001) was higher than in copepods and EPA was lower (p < 0.01). Furthermore, copepods had a higher ratio of phospholipids (PL) to total lipid (TL) than rotifers (p = 0.01). DHA, the lipid level measured as total fatty acids, all the vitamins analysed, as well as manganese, copper and selenium were higher or similar in rotifers than in copepods. The Artemia had a lower protein content than copepods (p < 0.001), similar to that in rotifers, and were characterized by taurine levels between those in rotifers and copepods (different to both, p < 0.01), high levels of cantaxanthin, no astaxanthin, and higher average levels of vitamin C, E and K (p < 0.01) compared to copepods. Iodine (p = 0.02) and zinc (p < 0.001) levels were lower in Artemia than in copepods. The fatty acid composition was characterized by similar DHA levels, but a much higher EPA:ARA ratio in copepods compared to Artemia. The level of total fatty acids and the ratio of PL to total lipids were similar to that in copepods.

Table 4 Nutrient levels in live food.

Leves of selected nutrient in the live feed used for cod larvae in (A) Exp-1 (mean ± SD, n = 2 for Artemia, n = 3 for rotifers and copepods) and (B) Exp-2 (n = 1 for Artemia , n = 2 for rotifers, n = 3 for small and large copepods; n.d., not detected). Superscripts indicate significant differences (ANOVA and Tukey’s post hoc test, p < 0.05).

A	Rotifers	Artemia	Zooplankton	
Dry matter (DM, g/100 g)	12 ± 1a	19 ± 2b	20 ± 2b	
Protein (g/100 g DM)	37 ± 4a	39 ± 0a	60 ± 2b	
Taurine (μ mol/g DM)	0.8 ± 0.3a	29 ± 1b	50 ± 6c	
Vitamins (mg/kg DM)				
A1	1.9 ± 0.1b	0.8 ± 0.3a	n.d.	
A2	n.d.	n.d.	n.d.	
C	553 ± 137b	617 ± 10b	94 ± 54a	
D3	0.7 ± 0.4	0.3 ± 0.0	n.d.	
E (α-tokoferol.)	333 ± 45b	280 ± 5b	105 ± 2a	
K	0.58 ± 0.24ab	1.59 ± 0.52b	0.21 ± 0.11a	
Carotenoids (mg/kg DM)				
Canthaxanthin	32 ± 4b	86 ± 28c	10 ± 1a	
Astaxanthin	11.1 ± 1.0a	n.d.	141 ± 15b	
Minerals (mg/kg DM)				
Iodine	2.6 ± 0.8a	3.9 ± 0.9a	35 ± 13b	
Manganese	6.9 ± 0.6	3.4 ± 0.7	6.5 ± 2.9	
Copper	10.6 ± 0.6b	9.5 ± 0.7ab	8.4 ± 1.0a	
Zinc	41 ± 6a	120 ± 14b	517 ± 75c	
Selenium	2.1 ± 0.1b	1.4 ± 0.1a	1.9 ± 0.3ab	
Lipids				
20:4n-6 ARA (% TFA)	1.9 ± 0.1c	2.4 ± 0.0b	0.6 ± 0.1a	
20:5n-3 EPA (% TFA)	4.4 ± 0.2a	4.1 ± 0.3a	10.8 ± 1.9b	
22:6n-3 DHA (% TFA)	32 ± 3a	19 ± 3b	22 ± 4ab	
EPA/ARA	2.3	1.7	18.0	
Sum fatty acids (mg/g DM)	98 ± 18	116 ± 15	95 ± 39	
Sum PL (% TL)	22 ± 2a	25 ± 0ab	29 ± 2b	
B	Rotifers	Artemia	Small copepods	Large copepods	
Dry matter (DM, g/100 g)	12 ± 2	5.11	13 ± 2	14 ± 1	
Protein (g/100 g DM)	38 ± 3a	55	58 ± 3b	67 ± 3b	
Taurine (μ mol/g DM)	0.7 ± 0.0a	40	59 ± 0b	63 ± 4b	
Vitamins (mg/kg DM)					
Thiamine	17.5 ± 4.7b	n.d.	5.0 ± 1.3a	4.7 ± 1.6a	
C	112 ± 104	142	144 ± 17	72 ± 39	
A1	n.d.	n.d.	n.d.	n.d.	
A2	n.d.	n.d.	n.d.	n.d.	
D3	0.20 ± 0.02	0.39	n.d.	n.d.	
E (α-tokoferol.)	671 ± 159b	802	58 ± 23a	84 ± 24a	
Minerals (mg/kg DM)					
Iodine	1.1 ± 0.2a	2.2	45 ± 9c	19 ± 5b	
Manganese	7.5 ± 0.6	3.5	93 ± 45	17 ± 4	
Copper	15 ± 3	14.5	28 ± 24	17 ± 4	
Zinc	52 ± 7a	157	413 ± 60ab	714 ± 160b	
Selenium	0.41 ± 0.19a	2.7	2.9 ± 0.5b	4.1 ± 0.3b	
Lipids					
20:4n-6 ARA (% TFA)	2.0 ± 0.1b	2.9	0.7 ± 0.1a	0.6 ± 0.3a	
20:5n-3 EPA (% TFA)	5.0 ± 0.0a	5.1	17 ± 3b	19.7 ± 0.4b	
22:6n-3 DHA (% TFA)	33 ± 1	9.1	35 ± 3	36 ± 2	
EPA/ARA	2.5	1.8	24.3	32.8	
Fatty acids (mg/g DM)	197 ± 16b	169	40 ± 2a	47 ± 2a	
Sum PL (% TL)	24 ± 2a	30	51 ± 7b	55 ± 5b	

The levels of selected nutrients in cod larvae fed rotifers/Artemia or copepods in Exp-1 were analysed at stages 3 and 5. Dry matter (DM) and protein did not differ between larvae fed the two diets (Table 6). At stage 3, taurin was almost 20 times as high in larvae fed copepods as in those fed rotifers/Artemia (40 and 2.1 µmol g−1 DM, respectively, p < 0.001). This difference became smaller, but was still present at stage 5 (46 and 39 µmol g−1 DM, respectively, p < 0.05). Vitamin levels were either similar in the two groups, or higher in larvae fed rotifers/Artemia than in those fed copepods. The same relationship applied to the minerals manganese, copper, zinc and selenium. Iodine was lower in larvae fed rotifers than in copepods at stage 3 (p < 0.05), but not at stage 5, where the groups had similar levels of iodine. The fatty acid composition at stage 3 partially mirrored the fatty acid composition of the feed, with a much higher EPA:ARA ratio in larvae fed copepods than in those fed rotifers (8.4 and 1.1, respectively, p < 0.01). Larvae fed rotifers had a higher DHA level than larvae fed copepods (37 and 31% of total fatty acids, respectively, p < 0.001). Although there were still statistical differences, the fatty acid composition appeared similar between the larval groups at stage 5. The lipid level measured as total fatty acids was similar in the two groups at stage 3, but higher in whole body of larvae fed rotifers/Artemia than in those fed zooplankton at stage 5 (p < 0.05). This was followed by a lower ratio of phospholipids to total lipid in cod larvae fed rotifers/Artemia in stage 5 (p < 0.01).

The differences in nutrient composition of the live feed in Exp-2 resembled those observed in Exp-1, where rotifers contained significant lower concentration of protein (p < 0.001), taurine (p < 0.001), iodine (p < 0.05) and zinc (p < 0.001) than copepods (Table 4B). The small copepod fraction is dominated by nauplii, the large copepod by juvenile and adult copepods (Table 5). In this experiment, the selenium level in rotifers was also lower than in copepods (p < 0.001), but above the requirement in fish of 0.3 mg kg−1 DM (NRC, 2011). There were also differences in fatty acid composition, where ARA was again higher and EPA lower in rotifers (p < 0.01). DHA was similar between the feeds. Copepods had a higher ratio of phospholipids to total lipids than rotifers (p < 0.01), corresponding to a lower total lipid level (p < 0.001). Vitamin A was not detected in any of the feed organisms, while vitamin D was absent in copepods. Vitamin E (p < 0.01) and thiamine (p < 0.05) were higher in rotifers than in copepods, while vitamin C, Mn, and Cu levels in the two feeds were similar. Only one replicate sample of the Artemia used in Exp-2 was analysed, so statistical treatment was not possible. However, the Artemia seemed to have more protein and taurine than in Exp-1, but still less than the copepods. The Artemia was also low in thiamine, vitamin A was not detected, while the content of other vitamins was similar to that in the rotifers. The analyses in Exp-2 indicate that iodine, manganese, and zinc levels were lower in Artemia than in copepods. The fatty acid composition was characterized by lower DHA and EPA levels in Artemia, but higher ARA levels. Lipid level and ratio of phospholipids to total lipid was similar to that in rotifers, namely with more lipid and a lower proportion of phospholipids to total lipid than copepods (Table 4B). With the exception of a higher Se content and a lower iodine content in large copepods, there were no significant differences in nutrient composition between small and large copepods, but the variation in some of the nutrients was wide (Table 4B).

Table 5 Proportion of zooplankton included in the analysis of live food in Exp-2.

	Dph/sampling date	Fraction (μm)	Copepod nauplii (%)	Juvenile + adult copepods (%)	Mussel larvae (%)	Polychaete larvae (%)	Mean DW/ind (μg)	
Small	5/13.04	80–150	98	2	0	0	10	
	6/26.04	80–150	87	10	3	0	7	
	21/9.05	80–150	53	5	34	8	9	
Large	5/13.04	150–180	64	36	0	0	7	
	6/26.04	180–250	0	100	0	0	30	
	21/9.05	180–250	0	100	0	0	17	

Table 6 Nutrient levels in cod larvae.

Exp-1: Levels of selected nutrients in cod larvae sampled at Stages 3 and 5. The group “Rotifers” were larvae first fed with rotifers and then after sampling at stage 3 the larvae were fed Artemia. Larvae in both groups were weaned to dry feed after sampling at stage 4. Mean ± SD, n = 3 tanks, superscripts in rows within each developmental stage indicate significant differences (t-test p < 0.05.).

Larval stage	3	5	
Group	Rotifers	Copepod	Rotifers	Copepod	
Age (dph) /Length (mm)	30/8.5	28/10.1	71/26.5	52/23.8	
Feed at sampling	Rotifers	Copepods	Dry feed	Dry feed	
Dry matter (DM, g/100 g)	32 ± 3	29 ± 3	26 ± 1	24 ± 1	
Protein (g/100 g DM)	70 ± 1	71 ± 1	67 ± 2	67 ± 2	
Taurine (μ mol/g DM)	2.1 ± 0.4a	40 ± 5b	39 ± 1A	46 ± 4B	
Vitamins (mg/kg DM)					
A1	6.2 ± 0.8	5.2 ± 0.8	6.7 ± 0.8	7.3 ± 1.4	
A2	0.6 ± 0.2	0.8 ± 0.1	1.9 ± 0.1B	1.3 ± 0.1A	
C	368 ± 52	395 ± 54	202 ± 10	192 ± 9	
D3	0.14 ± 0.00	0.05 ± 0.03	0.09 ± 0.02	0.07 ± 0.03	
E (α-tokoferol.)	95 ± 19b	54 ± 9a	103 ± 5B	44 ± 9A	
K	0.13 ± 0.07	0.04 ± 0.01	0.18 ± 0.02B	0.11 ± 0.03A	
Minerals (mg/kg DM)					
Iodine	0.8 ± 0.1a	2.0 ± 0.6b	1.7 ± 0.3	2.7 ± 1.0	
Manganese	3.9 ± 0.3b	2.1 ± 0.1a	5.7 ± 0.8	5.0 ± 0.3	
Copper	8.4 ± 0.5b	2.9 ± 0.2a	3.0 ± 0.2	3.2 ± 0.3	
Zinc	117 ± 6	120 ± 0	89 ± 4	85 ± 0	
Selenium	3.3 ± 0b	1.1 ± 0.0a	1.6 ± 0.1B	1.1 ± 0.0A	
Lipids					
20:4n-6 ARA (% TFA)	4.6 ± 0.1b	1.2 ± 0.0a	1.3 ± 0.0B	0.9 ± 0.1A	
20:5n-3 EPA (% TFA)	4.3 ± 0.1a	10.1 ± 0.1b	8.8 ± 0.1	9.0 ± 0.5	
22:6n-3 DHA (% TFA)	37 ± 0b	31 ± 0a	20 ± 0A	22 ± 1B	
EPA/ARA	0.9	8.4	6.8	10.0	
Sum FA (mg/g DM)	55 ± 7	65 ± 9	116 ± 5B	95 ± 8A	
Sum PL (% of TL)	51 ± 1	51 ± 1	37 ± 1A	42 ± 1B	

Discussion

This study shows that cod larvae fed copepods grow faster than larvae fed rotifers, and thus supports earlier studies (Busch et al., 2010; Busch et al., 2011; Koedijk et al., 2010). This differences in growth occurred in spite of recent improvements in rotifer nutritional quality, especially with respect to lipid and mineral enrichments (Hamre et al., 2013). It should be noted that during the first three weeks of feeding in Exp-1, there was no difference in growth between the groups, and that the greatest difference in growth rates occurred between 22 and 36 dph, i.e., when the intensive group was still being fed only rotifers. Also during the Artemia feeding period, the copepod group grew slightly better, but after weaning of the intensive group onto a formulated feed at 57 dph, the two groups grew at similar rates.

We suggest at least three possible reasons for the differences in growth rate: (1) Different availability of prey due to distribution or their behaviour in prey–predator interactions; (2) Different energy supply per ingested feed particle due to the small size of the rotifers compared to the increasing sizes of copepods given to larvae after 20 dph; (3) Differences in the nutritional values of the feeds.

There is a profound change in behaviour when rotifers are transferred into cold water, as they typically become immobile (and thus less attractive to visual predators, such as cod larvae) and gradually sink to the bottom (Fielder, Purser & Battaglene, 2000). In order to prevent sinking, our standard procedure is to acclimate rotifers to the same temperature as the larval rearing tanks prior to feeding, and to aerate the centre of the larval rearing tanks in order to keep the rotifers in suspension. Visual observations of the rearing tanks and of ingested feed in the intestinal tracts of the larvae verified that the distribution of the rotifers was constant over time and led to efficient feed intake by the larvae. High and immediate feed intake of rotifers has also previously been demonstrated for naive cod larvae at onset of exogenous feeding in a similar startfeeding system (van der Meeren, Mangor-Jensen & Pickova, 2007). It can therefore be concluded that the rotifers were highly available for the cod larvae.

With regards to the second hypothesis, the analysis shows that the energy content of copepods scaled with their size. The general pattern seems to be that when fish larvae are offered prey of different sizes, they choose the largest prey their mouth size can handle (Busch et al., 2009; Hunter, 1980; Olsen et al., 2000; van der Meeren, 1991), and this is also energetically the most advantageous. However, when the food is abundant, as it is under intensive farming conditions, sufficient energy may still be acquired from smaller prey consumed in larger numbers. An additional factor is that not only do larger prey items contain more energy per individual, but their surface to volume ratio also decreases, and large copepods thus contain relatively less indigestible chitinous exoskeleton. Therefore, since the rotifers have a fixed size that compares to the smaller copepod nauplii, the increase in prey size may also affect the nutritional composition if the cod larvae are given the opportunity to prey on larger copeopds. These questions were addressed in Exp-2. We found that the growth of larvae that were offered increasing size fractions of copepods or small copepod stages throughout the experiment had nearly identical growth rates. Larval dry weight was significantly, but only slightly larger at the end of the experiment for those fed the largest zooplankton. Both copepod groups grew far better than the intensive groups in both experiments.

In a similar setup that lasted only until 25 dph, Busch et al. (2011) observed a growth pattern similar to the present finding, with no significant differences between larvae fed small or large zooplankton and far better growth in both zooplankton fed groups than the cod larvae fed rotifers. This difference was only present at the end of the experiment. In their study the average length of the rotifers was 0.21 mm, small zooplankton 0.20 mm and large zooplankton 0.41 mm. This is slightly different from the zooplankton used in the present study in which the average total length of nauplii used was 0.15 mm, while the prosome length of the copepodids was 0.38 mm for the 80–180 µm filter fraction and about 0.5 mm for larger filter fractions. However, the range of prey sizes was large and also overlapped in both experiments. The observed growth patterns found by Busch et al. (2011) and in both of our experiments, supports the hypothesis that prey size in the range covered by our experiments and under intensive larval rearing conditions with high prey density, has only minor effects on larval growth. Therefore, the major cause of the differences in growth and development was most likely the differences in nutritional composition between zooplankton and rotifers/Artemia. In both experiments, the major growth difference started to appear during the later part of the period when the larvae were fed rotifers, but the difference remained during the period the larvae were fed Artemia. A similar growth pattern with equal growth until approximately 20 dph and thereafter higher growth rate in cod larvae fed copepods than in those fed rotifers, was also found by Koedijk et al. (2010).

The third hypothesis states that there are nutritional differences in the feed that underlie the differences in growth. We found major differences in the biochemical composition of rotifers and zooplankton in both experiments. In Exp-1, protein, taurine, carotenoids, iodine, and zinc were lower in rotifers than in copepods. The protein concentration in rotifers was similar to previous results at approximately 40 g 100 g−1 DM (Hamre et al., 2013; Srivastava et al., 2006), which represent only 2/3 of the concentration in copepods, and which is probably at or below the minimum requirement of cod juveniles (Åsnes, 2006). The nutrient composition of larvae was analysed at stage 3 when the rotifer/Artemia group was still fed rotifers, and at stage 5, where both groups had been weaned onto the formulated diet. Since the protein concentration in feeding animals is usually quite constant and only in extreme cases affected by diet, low dietary protein will usually result in reduced growth. This was confirmed by our study, since the dietary protein concentrations differed while protein concentrations in the larvae were similar. Thus, the lower rate of growth in the rotifer group at stage 3 may be explained by lower dietary protein. The protein concentration in Artemia used in Exp-1 was also low compared to that in copepods. The reason that the growth difference occurred first after 20 dph may be that the growth of muscle is relatively slow and characterized by recruitment of cells from 4 to 20 dph. Division of cells already present in muscle, along with growth of individual muscle fibres seem to speed up after 20 dph, resulting in faster muscle growth (Galloway, Kjørsvik & Kryvi, 1999) and perhaps resulting in a higher protein requirement, as suggested by Hamre et al. (2014).

Previous studies have already found high concentrations of taurine in copepods and very low concentrations in rotifers (Hamre et al., 2013; van der Meeren et al., 2008). Rock sole (Lepidopsetta polyxystra) larvae fed taurine-enriched rotifers (Hawkyard, Laurel & Langdon, 2014) had a whole-body concentration of taurine that was similar to the taurine concentration of copepod-fed larvae in the present study. In their experiment, larvae doubled their weight compared to control fish that were fed rotifers without taurine enrichment during a seven-week experiment. Several other studies have also shown that taurine may improve growth and development of marine fish larvae (Conceicão et al., 1997; Kim et al., 2014; Pinto et al., 2010; Pinto et al., 2013a; Pinto et al., 2013b; Pinto et al., 2012; Takeuchi et al., 2000; Takeuchi et al., 2001). Taurine is important for osmoregulation and bile salt production. Deficiencies may also result in lipid accumulation, mitochondrial damage and resulting oxidative stress, neurological anomalies and heart failure (Espe et al., 2008; Espe, Ruohonen & El-Mowafi, 2012; Jong, Azuma & Schaffer, 2012; Militante & Lombardini, 2004), suggesting that taurine deficiency may cause reduced growth in cod larvae that are fed rotifers.

Astaxanthin in copepods and cantaxanthin in Artemia may act as antioxidants (Stahl & Sies, 1999) or as pro-vitamin A compounds (Rønnestad et al., 1998; Moren, Næss & Hamre, 2002). In Exp-1, the vitamin A requirement of intensively reared cod larvae (Moren, Opstad & Hamre, 2004) was most probably covered by retinol in the rotifers, since the larval vitamin A level was similar to that in the copepod-fed group. Carotenoids were not analysed in the larvae of the present study, but visual examination showed that copepod-fed larvae displayed more colour than rotifer-fed larvae, as was also observed by Oie et al. (2015, unpublished data) and Busch et al. (2011). However, based on the well-known antioxidant effects of carotenoids (Stahl & Sies, 1999), it is unlikely that these differences would result in the large divergences of larval growth observed by us.

The concentration of iodine was considerably lower in the rotifers than in the copepods. However, the optimal level of iodine in rotifers has been estimated at 3.5 mg kg−1 DM (Penglase et al., 2013), which is within the range of variation of the levels in the rotifers analysed in the present experiment. It is therefore unlikely that iodine deficiency caused the reduced growth in rotifer-fed larvae.

Zinc is a necessary co-factor in the catalytic activity of more than 300 proteins involved in growth, reproduction, development and vision (Bury, Walker & Glover, 2003). In our study, the level of zinc in copepods was more than ten times as high as in rotifers. Despite this, the larval concentrations at stage 3 were similar for the two feeding regimes at approximately 120 mg kg−1, in line with the recognised tight regulation of zinc levels in fish, both at cellular and organism level (Bury, Walker & Glover, 2003). Penglase et al. (2013) fed cod larvae rotifers enriched with zinc to 87 mg kg−1 vs. 47 mg kg−1 in the control rotifers, without producing any detectable differences in growth. Combined with the present results and the zinc requirements of fish (15–37 mg kg−1, NRC, 2011), this indicates that rotifers in the present experiment had sufficient levels of zinc to promote growth in the larvae.

The fatty acid and lipid composition of rotifers and zooplankton displayed certain differences in both experiments. In Exp-1, ARA and DHA made up a lower, and EPA a higher, fraction of total fatty acids in rotifers than in copepods, and the proportion of phospholipids (PL) to total lipids was slightly lower in the rotifers. However, the differences in PL and fatty acid composition were probably not sufficient to explain the large difference in growth.

The concentrations of selenium, manganese, and copper, which previously have been shown to be lower in rotifers than in copepods (Hamre et al., 2008a), were at similar levels in the present experiment. All the other nutrients measured were present at higher or similar levels in the rotifers and copepods, and according to current knowledge, within the safe window of larval requirements. Apart from thiamine and the macrominerals, B vitamins were not analysed, since they have already been shown to be sufficient in rotifers, based on comparisons with copepods (Hamre et al., 2008a).

There were some important differences in the nutrient composition of live prey between Exp-1 and Exp-2. Iodine and selenium concentrations in rotifers were lower in Exp-2, but did not fall below the requirement given for fish (NRC, 2011). Furthermore, vitamin A was below the detection limit of the analytical method in all prey organisms in Exp-2. Artemia and copepods contain sufficient amounts of carotenoids to cover the vitamin A requirement of the larvae (Moren, Næss & Hamre, 2002), but in larvae fed rotifers in Exp-2, vitamin A may have been deficient. The lipid level in rotifers was higher than in copepods in Exp-2, and the proportion of PL to total lipid was therefore higher in the copepods. Finally, the concentration of protein in Artemia was higher in Exp-2 than Exp-1. The rearing temperature was also different between the experiments. Given all these differences in nutritional and physical conditions, the growth patterns were strikingly comparable between the experiments in this study, and to that observed by Busch et al. (2011). The growth in larvae fed rotifers may have been further reduced in Exp-2 due to a low vitamin A level, but the similarity in nutrient composition between small and large copepods, along with similar growth rates in the larvae fed on these, strongly suggests that prey size was of minor importance for the difference in growth observed in Exp-1.

In conclusion, differences in nutritional composition of prey are the most plausible explanation for the major divergences in growth and development of cod larvae fed natural zooplankton versus larvae fed enriched rotifers and Artemia. Low levels of protein and/or taurine in rotifers and Artemia are most likely the cause of poor growth in larvae fed rotifers. Since taurine is a water-soluble compound, large amounts are needed for rotifer enrichment, unless it is incorporated directly into particles that can be filtered out of the water (Hawkyard, Laurel & Langdon, 2014; Nordgreen, Hamre & Langdon, 2007). If poor growth is related to the low protein concentration in the rotifers, it might be more difficult to correct, since rotifers cannot easily be enriched with protein due to their own metabolism (Hamre et al., 2013). Possible suboptimal levels of zinc and carotenoids, ratio of PL/TL, and composition of fatty acids may also have contributed to slow growth in the cod larvae fed rotifers. However, these nutrients can be manipulated to a certain extent in rotifers (Hamre et al., 2013; Nordgreen, Penglase & Hamre, 2013; Olsen et al., 2014). The larval requirements for all nutrients in cod cannot easily be met by means of traditional protocols for enriching rotifers and Artemia. This require development of new protocols and adoption of new approaches to live feed enrichment during cultivation of early life stagesin cod and other marine fish.

Supplemental Information

Supplemental Information 1 Hydrography (Exp 1 & 2)

Click here for additional data file.

Supplemental Information 2 Length weight data (Ex 1)

Click here for additional data file.

Supplemental Information 3 Length weight data (Exp 2)

Click here for additional data file.

Supplemental Information 4 Prey added (Exp 1 & 2)

Click here for additional data file.

Supplemental Information 5 Nutrient composition of feed organisms and larvae (Exp 1)

Click here for additional data file.

Supplemental Information 6 Nutrients in feed organisms (Exp 2)

Click here for additional data file.

We thank the following technical staff at IMR Austevoll for their excellent assistance with the experiments; Stig Ove Utskot, Lillian Eggøy, Annhild Engevik, Margareth Møgster, Velimir Nola, Albert Rams, Michael Rejmer, Signe Lise Storebø and the technical staff at NIFES, in particular Kjersti Ask, the responsible technician for the nutritional analyses.

Additional Information and Declarations

Competing Interests

Author Contributions

Animal Ethics

Trina F. Galloway is an employee of SINTEF Fisheries and Aquaculture, and Kristin Hamre is an Academic Editor for PeerJ.

Ørjan Karlsen, Terje van der Meeren and Kristin Hamre conceived and designed the experiments, performed the experiments, analyzed the data, contributed reagents/materials/analysis tools, wrote the paper, prepared figures and/or tables, reviewed drafts of the paper.

Ivar Rønnestad conceived and designed the experiments, performed the experiments, wrote the paper, reviewed drafts of the paper.

Anders Mangor-Jensen, Trina F. Galloway and Elin Kjørsvik conceived and designed the experiments, wrote the paper, reviewed drafts of the paper.

The following information was supplied relating to ethical approvals (i.e., approving body and any reference numbers):

This study was carried out within the Norwegian animal welfare act guidelines, in accordance with the Animal Welfare Act of 20th December 1974, amended 19th June, 2009, at an facility with a general permission to conduct experiments involving all developmental stages of fish (code 93) provided by the Norwegian Animal Research Authority (FDU, www.fdu.no). The first trial was assumed to be a nutrition trial based on all available studies up to the date of the trial, no specific permit was required under the guidelines. The second experiment was approved by FDU, FOOTS ID 5448.

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
