# Peer review of "Copepods enhance nutritional status, growth and development in Atlantic cod (Gadus morhua L.) larvae — can we identify the underlying factors?"

_PeerJ, doi:10.7717/peerj.902_

## Round 0.1 · original submission · Major Revisions

Please make the changes suggested by the reviewers and pay special attention to improving all phrases that are ambiguous.

·

Basic reporting

The manuscript would benefit from language editing.

Experimental design

Experimental design is clear and appropriate. However, method description for biochemical analysis is not sufficient. Approach for statistical analysis is inappropriate and many of the tests are not applied correctly.
Ethical statement is missing.

Validity of the findings

Difficult to evaluate as data analysis is inadequate.

Additional comments

The study addresses role of food quality for fish larvae growth, which is an important issue in aquaculture, although without clearly formulated hypotheses it is not quite clear what relationships (and why) were evaluated. The experiments were executed in an accurate manner and the manuscript presents a valuable data set. However, the data analysis, statistics and synthesis are inadequate and do not provide a conclusive interpretation of the observed responses. In particular, the relative importance of various biochemical components in the feed for fish growth was not evaluated; the pairwise comparisons can provide only indicative evidence for possible effects of the different diets. Moreover, some biochemical constituents of the diets may co-vary, which means that not every component that differ between the preys would be really responsible for the growth response. I would suggest to apply regression analysis or multivariate analysis aiming at detecting the most influential predictors, including effect size analysis. PLS regression with VIP scores could be an alternative method. This would provide an evaluation of the effects of specific nutritional variables on specific growth variables. Formulating hypotheses and re-analyzing the data to match the hypotheses implies a major revision of the study and this would be my recommendation.

Some specific comments:

Abstract: Confusing description of the experimental treatments and time points at which observations/measurements were conducted. Also, it is unclear – what were the “other measured nutrients” that “were similar or higher in the rotifers”. Concluding sentence is missing.

Intro. Hypotheses and research questions are needed, otherwise, it is very fuzzy what and why was tested.
Lines 66-67: Rephrase the sentence.
Lines 69-70: “Prey and larval nutritional status were investigated during the experiment” Perhaps – monitored, not investigated?
Lines 70-72: “Additional larval samples were taken for assessment of the ontogeny… transcriptomics, microRNA-sequencing and metabolomics, which is reported elsewhere”. Unclear – why this is relevant for the data interpretation in the current study?
Lines 75-76: “In addition to the nutrient composition, the size and energy content of the rotifers and Artemia may limit growth”. This paragraph should follow the paragraphs about the differences in nutrient composition (lines 41-47 and 49-64).

M&M. The description of the rearing system is unnecessary long and detailed. I would suggest to summarize this information as a table. Much of the information presented in the figures is repeated in the text. By contrast, virtually no rationale and methodological details are provided for the biochemical analyses, which makes it difficult to follow the results. For example, PL/NL ratio – why was it used, what was your hypothesis regarding this aspect of food quality and how exactly was it measured? Other biochemical variables need the same kind explanations as well, which could be perhaps summarized in a table. Finally, in the statistical analysis, Exp-1 and Exp-2 should be treated as variables when comparing treatments that are identical in these two experiments.

Results. Too much details of secondary importance, which makes the results section difficult to follow. Parametric tests for survival data (%) are unappropriated, because it is most likely not normally distributed and, therefore, should be treated as such. The same would also apply to other data expressed as % or ratio. Altogether, more graphical illustrations would help to see the differences between the treatments/experiments. Why identical treatments in Exp-1 and Exp-2 produced different results? (They seem different but this was not addressed at all).

Discussion. The results interpretation is weak, descriptive and shallow, but this is because of the absence of the clear hypotheses and inadequate data analysis. Re-structuring the paper and harmonizing hypotheses with research methods and conclusions would make it clearer and more understandable, but also more valuable in terms of the establishing specific dietary components as cod growth predictors during different growth periods.

·

Basic reporting

L327-336: You could use some more detail in the statistics section. You have several results where you give one p-value for comparisons over multiple time points. Please elaborate about how you pooled your data?

Very minor point: I don’t really like astaxanthin and castaxanthin to be referred to as pigments (L530). While they are pigments, they also serve a variety of important functions in fish (as you noted on line 552). I would suggest listing them by name.

I’m also not a big fan of referring to the larvae fed rotifers and copepods as “intensive”. I think this implies that the copepod groups were “extensive” which isn’t really true in this case. This is another very minor point, but you may reconsider this label. Perhaps “cultured prey” would be a better name?

On line 534, you suggest that protein is determined by the genetic code but then go on to imply that it is impacted by dietary protein. You may want to explain this a bit more clearly, especially the genetic component. Furthermore, you did not see any differences in total protein levels in the larvae, please elaborate.

Figures: I printed this out in Black and White, so some of my figure issues may have been due to that. However:

Figure 1: In B&W, this is hard to read. It is difficult to tell which line is temp and which is O2.

Figure 2 and 3: I think this would be better expressed in terms of number of prey per volume (i.e. liter or ml). Specifically because the tank sizes were different between the two experiments so they are hard to compare. Also, number of prey per volume is a bit more universal.

Figure 4: In B&W it is impossible to tell the two groups apart.

Figure 5: These are nice figures. However, if you change your method to a regression analysis (see “Experimental design”) then I think you should use the linear estimate rather than the connected-line format.

Experimental design

1. (L274) I don’t know why you sampled the larvae 2 hours after feeding. I would imagine this means that they had full guts. Meaning that the gut contents contributed to the larval nutrient concentrations as well as the dry weights of the larvae. Please justify?
2. I don’t think you had enough samples of live prey to make statistical comparisons see “Validity of findings”.
3. Length and weight comparisons in EXP-1: I am not entirely clear why you sampled larvae on different days. From a nutrient composition standpoint, it makes sense to compare larvae from similar stages since these concentrations are influenced by ontogeny. However, I think lengths and weights are better compared at similar ages or degree days. This is especially problematic since your “stages”, by your definition, were derived from larval lengths (L364). So if I understand correctly, you were trying to compare the differences in larval lengths at similar larval lengths. You may be able to strengthen this comparison by using regression analyses, rather than ANOVAs, to analyze length and weight data. This would eliminate the problem of different sampling dates. Also, it is unclear how the comparisons were made over time-ranges, for example from day 4 to 22 (L369). If you were comparing average lengths over long periods (i.e. data was pooled for multiple sampling dates), you may have received false positives. Again, this may be remedied with regression analyses. Alternatively, you may choose to perform statics on endpoint data or some other summary statistic.
4. Length and weight comparisons of EXP-2: You have sampled the same tanks over time and are conducting ANOVAs at each time point. This is a repeated-measures problem since data were tied together by tank. You may want to consider a repeated-measures (or similar) model to correct for this problem. Unfortunately, this is actually very common in aquaculture studies. Again, you could solve this by using end-point data or a summary statistic.

Validity of the findings

The major weakness of this manuscript is that there are not enough samples (n<3) of rotifers and Artemia to make strong statistical conclusions about nutrient differences. As noted by the authors, the nutrient composition of Artemia cannot be compared with rotifers and copepods due to the low sample sizes. For instance, n=2 and 2 for Artemia in Exp-1 and Exp-2, respectively. In EXP-2, “n” is also small for both rotifers and small copepods. Given that this paper is focused on nutritional differences between these organisms, I think that this is a big problem. Is it possible to pool the data from both experiments, especially for the rotifers and Artemia? I believe that the enrichment and sampling protocols were similar in these trials. If so, you may gain statistical power by using the combined data. The problem with low sample sizes is that you may have false negatives in your data set (Type 2 error). In other words, there may be other nutrients that were deficient for the larvae. Of course, if you are able to combine the data, you will need to analyze with ANOVA rather than T-tests.

Additional comments

This manuscript describes a set of experiments where rotifers and Artemia are compared with copepods in laboratory bases growth trials with cod larvae. I believe that this work is very relevant to the area of larval fish nutrition. Identifying the key nutritional differences between these organisms is an important step forward in this area. Furthermore, this paper is logically written and well thought out. However, I think that the experimental design with respect to the nutritional analysis of the live prey is very weak primarily due to small sample sizes. Since this is a fundamental component of the manuscript, I will recommend this paper for major revisions. I have also included a number of comments for improvement and disambiguation.

Reviewer 3 ·

Basic reporting

No comments. The basic aspects of this manuscript are in order.

Experimental design

There are part of the Experimental Design that need to be elaborated by the authors. See General Comments to the authors.

Validity of the findings

There are aspects of use of statistical test that need the attention of the authors. See under General Comments for the Author.

Additional comments

PeerJ #2994 – Copepods enhance nutritional status, growth, and development in Atlantic cod (Gadus morhua L.) larvae – can we identify the underlying factors? by Karlsen et al.

This is an interesting study where the authors have investigated several aspects of larval rearing of cod fed natural zooplankton, rotifers and Artemia. The underlying problem is that cod first-fed on enriched rotifers and Artemia have inferior development and growth compared to groups first-fed natural zooplankton.
To shred a light on the underlying factors behind this the authors conducted two experiments. In the first experiment (Exp-1) groups of cod larvae were fed either a) natural zooplankton, mainly copepods, increasing the size of prey as the larvae grew or b) enriched rotifers followed by Artemia (the intensive group). In the second experiment (Exp-2) two groups of larvae were fed as in Exp-1, while a third group was fed copepod nauplii (approximately the size of rotifers) throughout the larval stage.
The aim of the first trial was to look at developmental differences between groups of cod larvae fed zooplankton vs. enriched rotifers and Artemia, whereas the aim of the second study was to investigate the possible effect of prey size on development and growth.

The manuscript is clearly written although not particularly novel. Experiments similar to Exp. 1 in this manuscript have been published elsewhere and are cited in this manuscript. Exp. 2 is, however, novel in its approach do look at prey size specifically.

General comments
1. Several similar works have been published elsewhere so it is difficult to see the need for the extensive description given in the M&M section (almost 11 pages). E.g. there is a 3 page description of the Start Feeding System and a 3 page description of Larval feed and feeding regimes. There is clearly a room for extensive rewriting of this and other sections of this Chapter.
2. But with almost 11 pages one would expect a thorough description of the Experimental Set-up but this basic part is actually missing. Instead one has to read through the whole M&M to understand the set-up of the experiments. The authors are strongly encouraged to write separate Experimental Set-up chapters (one for Exp. 1 and one for Exp. 2).
3. The statistical section is very confusing. In both trials replicates were used so the basic set-up is a nested set-up. In the stat chapter in the M&M it is stated that the continuous variables investigated (length, weight) were compared with two-way ANOVA (treatment, time) followed by Tukey HSD post hoc test. This is incorrect as three-way nested ANOVA should be applied in the start. If significant a reduced model could be applied. If the authors want to include both treatment and time an interaction between those variables (or the absence of if) must be described in the Results. This is currently missing.
4. The authors use t-test for analysing SGR, DLI and nutrient composition of feed and larvae. Again the fact that the authors have used replicates is ignored with the use of t-test. SGR and DLI results for Exp. 1 and Exp. 2 are given in Table 3a and 3b, respectively and here Tukey HSD post hoc test is used to analyse possible difference (presumably an ANOVA is used prior to this). So here there is a mismatch between M&M and actual test used. In general the use of t-test is not warranted given this experimental set-up.
5. In Table 2 nested ANOVA is listed as the method used, but not in the M&M description. For Table 4 no information is given about the statistical test used for analysing Exp. 2.
6. The different trials described in this manuscript (Exp.1, Exp. 2) differ in their experimental set-up in two fundamental ways.
a. Temperature. In Exp. 1 the rearing temperature was around 11.6°C, whereas in Exp. 2 the experimental temperature was not adjusted but varied between 7.1 and 8.6°C. Growth and development of cod larvae will be very different at these two temperatures (e.g. Fig. 1 of this study). Temperature will be confounding effect and any comparison between the two trials will be hampered by this confounding effect. This is very unfortunate as it means that comparison between Exp. 1 and Exp. 2 should be done with extreme care.
b. Experimental length. Exp. 1 was terminated at day 84 ph, whereas Exp. 2 ended 47 dph. This is the second confounding variable in this set-up.
7. Accordingly any comparison between Exp. 1 and Exp. 2 must be done with extreme care. With so fundamental different experimental set-up and confounding effect it might perhaps be best that these trials are published separately.
8. The oxygen level in both groups in Exp. 1 drops below 75%, but the drop is more abrupt in the copepod group. The possible effect of this should be described and discussed. Why are the oxygen levels from Exp. 2 not shown?
9. Different genetic groups were used in Exp. 1 and Exp. 2 (lines 93 vs. 120).

Specific comments

- Table 5. The time format is very strange (13.apr). Perhaps Norwegian?
- Could the authors elaborate why the growth is similar in both experiments prior to 22 dph?
- The numbers of larvae used in the different analyses is very difficult to follow and some cases this information is not given. May I kindly ask the authors to include an overview of the numbers of larvae used in the different analyses!

In summary, I have to admit that although the topic is truly interesting the experimental conduct has some weaknesses that the authors have ignored in the present version of the manuscript. The basic aspects of this manuscript are in many ways solid, but weaknesses of the set-up must be dealt with in an adequate manner before this manuscript can be published. I would recommend that this manuscript could be reconsidered pending major revisions. The shortcomings of the experimental set-up (temperature, experimental length, oxygen levels etc) and related findings will have to be acknowledged. The statistical treatment is not correct and must be changed. The authors must be consistent in their use of statistical methods as there is a discrepancy between description given in the M&M and methods used in the Results section.

---

## Round 0.2 · accepted · Accept

Thank you for revising the initial manuscript according to the reviewer suggestions.